# Design of Cereal Products Naturally Enriched in Folate from Barley Pearling By-Products

**DOI:** 10.3390/nu14183729

**Published:** 2022-09-10

**Authors:** Stefania Ruggeri, Elisa De Arcangelis, Altero Aguzzi, Maria Cristina Messia, Emanuele Marconi

**Affiliations:** 1Council for Agricultural Research and Economics (CREA), Research Centre for Food and Nutrition, Via Ardeatina 546, 00178 Rome, Italy; 2Department of Science and Technology for Humans and the Environment, Università Campus Bio-Medico di Roma, Via Alvaro del Portillo 21, 00128 Rome, Italy; 3Dipartimento Agricoltura, Ambiente e Alimenti (DiAAA), Università degli Studi del Molise, Via Francesco de Sanctis, 86100 Campobasso, Italy

**Keywords:** barley, biscuits, folate, folate enriched cereals, pearling, pasta

## Abstract

Folate is a fundamental vitamin for human health in prevention of many diseases; however, unfortunately its deficiency is widespread, so a greater availability of folate rich foods is desirable. The aim of this study was to design new cereal products naturally enriched in folate using barley flour from pearling as ingredient. Folate content of unfortified and fortified commercial grain-based products was considered to identify the best ingredients for new formulation and for folate content comparisons. Nineteen Italian barley cultivars were evaluated for their folate content and Natura was chosen for its highest folate levels = 69.3 μg/100 g f.w. Application of pearling gave a by-product flour with a high folate level: 221.7 ± 7.0 μg/100 g; this flour was employed to design pasta and biscuits naturally enriched in folate: 87.1 μg/100 g and 70.1 ± 3.7 μg/100 g f.w., respectively. Folate content of new products is higher than commercial samples: 39.2 μg/100 g in refined pasta, 60.4 μg/100 g in wholemeal pasta, 62.1 μg/100 g in fortified biscuits and 10.4 μg/100 g in unfortified ones. Enriched pasta had higher folate retention (68.5%) after cooking compared to the fortified one (27.8%). This research shows promising results concerning the pearling technique to design new cereal products naturally enriched in folates.

## 1. Introduction

Folate is a crucial vitamin for human health. Many studies have demonstrated that an adequate folate status preserves from megaloblastic anaemia and, during the preconceptional period, significantly reduces the risk of many Adverse Reproductive Outcomes such as Neural Tube Defects (NTDs) [1], cleft lips, some cardiac congenital defects [2,3], intrauterine growth restriction, preterm birth [4]. Folate deficiency can cause hyperhomocysteinemia, a risk factor for cardio-cerebrovascular diseases [5] associated also with cognitive decline and some forms of dementia, as well as osteoporosis [6,7]. Systematic reviews of the literature and clinical studies have highlighted also a possible dose-dependent inverse relationship between folate intake and the risk of some forms of cancers such as: pancreas, oesophagus, stomach, prostate, bladder [8,9]. Unfortunately, folate deficiency occurs in many countries (i.e., Ireland, Italy Norway, Sweden, Netherlands, Greece and United Kingdom), and several studies showed low folate plasma values in adults, women of a childbearing age and people aged ≥50 years, highlighting a deficiency situation in many population subgroups [10,11,12,13,14,15,16,17].

To improve folate status at population levels, some countries (i.e., USA, Chile, Australia) have long established mandatory folic acid fortification of cereal flours. Notwithstanding the success of these initiatives in reducing NTDs [18,19], some authors stressed the possible risks related to an excessive intake of synthetic folic acid over the upper level of 1000 μg [20] in masking vitamin B_12_ deficiency and the occurrence of unmetabolized folic acid in plasma whose outcomes on the human body have not been clarified yet [21,22,23,24]. Discrepant results stand out on this topic, leading concerns whether mandatory folic acid fortification should be introduced or not in other countries. In European countries nowadays only voluntary food fortification is allowed [Reg. (EC) No 1925/2006] and fortified products with folic acid are mainly featured by cereal-based products, i.e.: pasta, biscuits, crisp bread and other bakery products [25].

On the basis of these considerations, the development of new products naturally enriched with folate would be suitable to introduce alternatives to fortified products, mainly in countries where mandatory fortification is not allowed.

Among cereals, barley (*Hordeum vulgare* L.) is a good source of folate [26] that is not uniformly distributed in cereal kernels, more concentrated in the outer layers such as pericarp, testa, aleurone layer, and germs [27]. Barley is a cereal that draws attention for its health properties, being a source of bioactive compounds such as β-glucans ((1 → 3), (1 → 4)-β-D-glucan) [28] which can have beneficial effects on cholesterol and sugar levels [29,30]. Futhermore, barley is a very interesting crop: it is cultivated globally for both animal feed and human consumption, holds the fourth position among world cereals production [31], fits very well to different environmental conditions; compared to wheat, it matures earlier thus requiring a lower amount of water, while concurrently demanding for less nitrogen fertilizers to reach the highest yield [32].

Many techniques were applied to obtain barley fractions enriched in folate and in other bioactive compounds, and pearling seems to be the most promising [27,33,34,35,36].

Pearling process consists of an abrasion of cereal grain to remove the outer layers and is normally applied to barley for food uses: pearled barley is usually employed in traditional recipes, as a rice substitute or as a breakfast cereal [37], while pearling by-products are conveyed into feed processing.

The aim of this study was to design new cereal-based products—pasta and biscuits—naturally enriched in folate using barley flour from pearling by-products as an ingredient. The nutritional potential of new experimental products was assessed in terms of folate content and compared with fortified/non-fortified grain cereals foods already on the Italian market.

## 2. Materials and Methods

### 2.1. Barley Samples

Nineteen barley (*Hordeum vulgare* L.) genotypes were grown in the same experimental field in the South of Italy and obtained from Agroalimentare Sud S.p.A. (Melfi, (PZ), Italy). They were collected and stored at +4 °C. Before analysis, samples were dehulled and milled using a laboratory mill mod. IKA A 10 (IKA-WERKE GmbH & CO. KG, Staufen, Germany) using a water-based cooling system.

### 2.2. Commercial Samples of Conventional and Fortified Cereal Products

Four different brands of “00” type soft wheat flours, four brands of “0” type soft wheat flours, four brands of commercial semolina, five brands of durum wheat semolina pasta, five brands of durum whole wheat semolina pasta, one brand of whole-meal pasta fortified with folic acid, four brands of shortbread biscuits and two brands of shortbread biscuits fortified with folic acid were purchased in local supermarkets and sampled in two different years (three packages for each brand per each years) from among the most consumed brands [38]. Type “00” and type “0” soft wheat flours are product categories that differ in their refining rate, according to the legal requirements set by the Italian Presidential Decree No 187/2001 (Type “00”: moisture = max 14.50%, ash content = max 0.55% d.m, protein content = min 9% d.m.; type “0”: moisture = max 14.50%, ash content = max 0.65% d.m., protein content = min 11% d.m.). Wheat flour samples from the same years were pooled (300 g from each package) and mixed in a laboratory; biscuit samples were homogenised, pooled and freeze-dried in a VirTis Genesis 25SES Pilot Lyophilizer (VirTis Co. Inc., Gardiner, NY, USA). The freeze-dried samples were ground with a refrigerated laboratory mill (model IKA A10-IKA Werke GmbH & Co. KG, Staufen, Germany) and stored at −20 °C before analysis.

### 2.3. Pearling

De-hulled barley cultivar Natura underwent a single pearling process (three-steps) according to an industrial flow chart. A Carter disk separator was firstly used to remove broken kernels and foreign grains, then samples were passed through a calibration disk to obtain uniform-sized grains (≥2.5 mm caliber). The process consisted in three pearling steps, recovering pearled barley kernels (PK1, PK2, PK3) and pearling by-products (BP1, BP2, BP3). The first step removed 14.7% of the initial kernel weight, the second step 19.3% and the third step 31.6%. The pearled barley kernel from each pearling step was milled in a laboratory mill mod IKA A 10 (IKA-WERKE GmbH & Co. KG, Staufen, Germany) with a water-based cooling system.

### 2.4. Gluten Extraction

Vital wheat gluten was produced according to Verardo et al. [39]. Briefly, durum wheat semolina (2 kg) and 1.2 L water were mixed in a dough mixer (NAMAD, Roma, Italy) at room temperature for 6 min (95–140 rpm). The dough was allowed to rest and then sieved (pore size 420 μm) and after 30 min of washing, the gluten fraction was dried under vacuum at <50 °C. The dried gluten was then milled (150–200 µm) and stored at room temperature before use.

### 2.5. Pasta Making

Selected commercial semolina (61%), barley pearling by-product (BP2) (35%) and vital wheat gluten (4%) were used to make pasta (spaghetti shape, two replicates) (Table 1) naturally enriched in folate, using an experimental pasta-making apparatus. The dough moisture was increased by 30% using tap water (30 °C) and mixed for 15 min (pressure vacuum 0.8–0.9 Bar, speed 30.4 rpm). After extrusion (spaghetti, diameter 1.7 mm), product was desiccated at high temperature 80–90 °C for 7.5 h [40].

### 2.6. Pasta Cooking

Pasta was cooked according to ISO 7304-1:2016 (ISO, 2016) in tap water (pasta:water ratio of 1:10) at the optimal cooking time (OCT). The optimum cooking time was taken at the moment that the white core of the pasta disappeared when squeezed between two test glasses. After cooking, samples were freeze-dried in a VirTis Genesis 25SES Pilot Lyophilizer (VirTis Co. Inc., Gardiner, NY, USA). Freeze-dried cooked samples were grounded with a refrigerated laboratory mill (model IKA A10-IKA Werke GmbH & Co. KG, Staufen, Germany) and stored at −20 °C before analysis.

### 2.7. Biscuit Making

Enriched biscuits were produced (two replicates) in a pilot plant by mixing commercial type “0” flour (45%) and pearling by-product (BP2) flour (55%) with sugar, butter, milk, eggs (Table 1). Ingredients were kneaded in a spiral kneader for 30 min. Dough was laminated, shaped and baked at 180 °C for 15 min in a rotary oven (CIMAV, Villafranca VR Italy). Samples were freeze-dried in a VirTis Genesis 25SES Pilot Lyophilizer (VirTis Co. Inc., Gardiner, NY, USA).

### 2.8. Moisture Analysis

For moisture determination, milled samples were dried according to ICC method 110/1 and analysed in triplicate.

### 2.9. Enzymes Preparation for Folate Determination

#### 2.9.1. Conjugase

Hog kidney conjugase was prepared according to Gregory et al. [41] from conjugase hog kidney acetone powder, porcine, type 2 k7250-10G (Sigma Chemical Co., St. Louis, MO, USA). Endogenous folate in hog kidney conjugase solution was subtracted in the final calculation of folate content in all samples.

#### 2.9.2. α-Amylase

0.5 g of α-amylase (EC 3.2.1., A-6211, Sigma Chemical Co., St. Louis, MO, USA) was suspended in 25 mL of distilled water. This solution was mixed (1:10) with activated charcoal and gently vortexed on ice for 20 min. No folate was detected in α-amylase solution after the microbiological assay with *Lactobacillus rhamnosus*.

#### 2.9.3. Protease

0.05 g protease (EC 3.4.24.31, P-5147, Sigma Chemical Co., St. Louis, MO, USA) was suspended in 25 mL distilled water. No folate was detected in the protease solution after the microbiological assay with *Lactobacillus rhamnosus.*

### 2.10. Folate Determination

For folate determination, samples (2.5 g) were extracted with 50 mL of 50 mM Ches (*N*-Cyclohexyl-2-amino ethanesulfonic acid) and 50 mM Hepes (4-(2-hydroxyethyl)-1-piperazineethanesulfonic acid) buffer containing 20 g L^−1^ sodium ascorbate and 10 mM 2-mercaptoethanol (pH 7.85), under nitrogen atmosphere and subdued light to prevent folate oxidation. Capped tubes were placed for 10 min in a boiling water bath, then cooled in ice, and centrifuged as reported by Pfeiffer et al. [42]. The extraction was performed in triplicate.

After extraction, trienzyme treatment was performed on samples according to Ruggeri et al. [43]. Thus, 25 mL of each extract was deconjugated at pH 4.9, with 1 mL of hog kidney conjugase solution for 3 h at 37 °C, under nitrogen atmosphere in a shaking water bath. Together with conjugase, 1 mL of α-amylase solution was added. After treatment with conjugase and α-amylase, samples were boiled for 10 min to inactivate the enzymes and pH was brought back to 7.0 with NaOH 0.2 M.

2 mL of protease was then added to each sample, and the enzyme was left to act for 1 h at 37 °C. At the end of the protease treatment, samples were heated in a boiling water bath for 10 min and centrifugated (12,000 rpm, 15 min, 5 °C). The residues were suspended into 5 mL of 0.1 M phosphate buffer pH 7.0 containing 0.2% ascorbate and recentrifuged for 10 min. Portions of samples were flushed with nitrogen and stored at −20 °C until the analysis.

Total folate was determined by a microbiological method [44] using *Lactobacillus casei* subsp. *rhamnosus* (ATCC 7469 NCIMB 6375) with growth medium FAAM (Folic Acid Assay Medium n F5422 Sigma Chemical Co., St. Louis, MO, USA), with some modifications.

In each assay, a calibration curve from 0 to 8 ng per folic acid tube was prepared using an aliquot of Pteroylglutamic acid (PGA) stock solution. Samples and standard solutions were inoculated for 18 h for *Lactobacillus* growth at 37 °C and the absorbance was read at 630 nm. To assess accuracy and repeatability of measurements, the Certified Sample “Wholemeal flour” CRM BCR-121 (Institute for Reference Materials and Measurements, Geel, Belgium)—folate content = 51.3 ± 7.1 µg/100 g—was analysed with each set of samples. The applied methods for folate determination showed very good accuracy with a Relative Error = 1.02 and good repeatability calculated as Relative Standard Deviation = 5.1% for the Certified Reference Material CRM 121. Folate retention (FR) after cooking in both new pasta and folic acid fortified pasta were calculated on as: FR (%) = cooked pasta folate content (dry weight)/uncooked pasta folate content (dry weight).

### 2.11. Statistical Analysis

A one-way analysis of variance (ANOVA) employing the Kruskal–Wallis non-parametric test at a significance level of 5% was carried out to determine significant differences in total folate content processed using the SPSS statistical software (version 22, SPS, Chicaho, IL, USA).

## 3. Results and Discussion

### 3.1. Folate Contents in Barley Cultivars

In order to identify the barley cultivar richest in folate, a screening of total folate content in nineteen Italian dehulled barley cultivars was carried out (Table 2). The folate content was affected by genotype, ranging from a minimum of 39.8 ± 2.7 μg/100 g d.m. in *Gothic* cultivar to a maximum of 78.9 ± 7.3 μg/100 d.m. in *Natura* cv, with a coefficient of variation of 14.6%. Italian dehulled Trasimeno and Ketos barley cvs showed 73.2 µg/100 g d.m. and 65.3 µg/100 g d.m. folate content respectively, while higher folate levels such as 103.3 µg/100 g d.m. were detected in the hulless Mona cultivar [34]. Edelmann et al. [27] analyzed folate content of five hulled Finnish barley cultivars in different years, with value ranging from 62.5 to 91.8 µg/100 g d.m. These values are slightly higher than the ones reported by Andersson et al. [45] in ten barley genotypes grown in Hungary in 2004–2005, with folate values amounts from 51.8 to 78.9 µg/100 g d.m.

Other studies of the last fifteen years showed a great variability in folate content also in other cereals grains as well, such as wheat and rice, related to the type of cultivar and storage. Riaz et al. [46] analysed 315 wheat genotypes grown in North China and detected folate content ranging from 10.1 to 91.4 µg/100 g grains. Piironen et al. [47] in HEALTHGRAIN project found folate range from 36.4 to 77.4 µg/100 g d.m. in winter wheat and from 32.3 to 74.1 µg/100 g in spring wheat. Liang et al. [48] underlined that six–eight months storage of wheat caused a folate loss of 26%.

Concerning rice, folate values ranged from 13.3 to 111.4 μg/100 g in brown cultivar, whereas milled rice varied from 10.3 to 77.7 μg/100 g, with a folate loss after storage of 23% [49]. Many of these studies demonstrated that folate content in cereal grains may range a lot, mainly in relation to cultivar and storage.

### 3.2. Pearling Process to Obtain Barley Fractions with a High Folate Content

Table 3 shows the folate levels in pearled caryopsis and in derived pearling by-products of Natura cultivar (folate content = 69.1 ± 6.4 µg/100 g f.w.). Pearling by-products were characterised by a high folate content: in the first pearling flour (BP1), folate levels were more than 2-fold higher (173.5 μg/100 g f.w.) than the folate content of the initial caryopsis. In the second pearling process (BP2, yield = 19.3%) folate, was more concentrated, reaching the levels of 221.7 ± 7.03 µg/100 g f.w. (more than threefold higher than initial caryopsis); folate content decreased in the third pearling by-product (BP3, yield = 31.6%), with values of 177 ± 16.9 μg/100 g. Enrichment in folate was gained with the second pearling by-products (BP2), where likely aleuronic layer cells and the germ—very rich in folate—are concentrated, while the reduction in folate content found in the third pearling by-products (BP3) is due to a higher presence in this layer of starchy endosperm cells, which are less rich in folate [27].

Giordano et al. [34] obtained a similar folate enrichment (threefold) in a pearling by product: 464.7 µg/100 g d.m. after processing a hull-less barley cultivar Mona with higher folate content as 103.3 µg/100 g d.m. compared to our Natura cv.

Blandino et al. [50] found the highest content of folate in germ fraction: 85.1 µg folic acid equivalents/100 g d.m. after dry-degermination system established of a whole grain maize hybrid for animal feeding (total folate content of 35.8 µg folic acid equivalents/100 g). Fenech et al. [51] determined about 500 μg/100 g in aleurone flour.

The vast differences in the amount of folate among these studies are explained by different cereal grains at the start, differences in milling and particularly by the ratio of folate–rich aleurone layer in the fractions [26,48,52].

### 3.3. Folate Content in Unfortified Commercial Semolina and Wheat Flours

To identify semolina and flour very rich in folate, for the formulation of new cereal products, four brands of commercial unfortified semolina, four brands of “00” wheat flours and four brands of “0” wheat flours were examined for their folate content (Table 4). In commercial semolina samples, folate content ranged from 35.0 μg/100 g f.w. (brand 1) to 40.2 μg/100 g f.w. (brand 4) and the latter was used for production of new pasta (Table 1). High levels of folate were detected in “0” flours due to the presence of bran fraction with very high folate contents, till 69.3 ± 1.2 μg/100 g in brand 4 (Table 4) and therefore this sample was chosen for biscuit making. Overall, the average folate value in “00” flour was 38.8 μg/100 g, and all “00” samples showed lower values (*p* < 0.05) compared to “0” flours. Very few studies were conducted on folate content of wheat flours and no data—that the authors are aware of—are available on semolina. Arcot et al. [52] found 41 μg/100 g of folate in a sample of wheat flour, and Liang et al. [48] detected very low folate content: from 9.29 ± 0.96 μg/100 g and 10.64 ± 1.02 μg/100 in some wheat flours.

Rader et al. [53] in their study for the evaluation of the efficacy of mandatory folic acid fortification in USA, found folate levels in unfortified wheat flours about 88 μg/100 g.

### 3.4. Folate Content of New Cereals Products Naturally Enriched in Folate

Folate content was analysed in new cereal products naturally enriched in folate—pasta and biscuits—in comparison with commercial unfortified samples, as well as fortified ones (Table 5). Uncooked enriched pasta showed folate values of 87.1 μg/100 g, with higher level compared to unfortified pasta: commercial durum pasta samples ranged from 36.7 to 44.3 μg/100 g (mean value = 39.2 μg/100 g) and commercial durum whole wheat pasta ranged from 50.5 to 61.1 μg/100 g (mean value = 60.4 μg/100 g). On the contrary, fortified pasta with folic acid (uncooked) showed higher folate content, at 200 μg/100 g, compared to folate in the naturally enriched folate pasta examined. After cooking, enriched pasta maintained good folate levels, 26.1 ± 1.1 μg/100 g, higher in comparison with durum wheat pasta folate (mean value: 12.3 μg/100 g; range: 7.6–15.7 μg/100 g) and whole wheat pasta (mean values: 16.5 μg/100 g; range 10.5–18.4 μg/100 g), and slightly lower than folate in pasta fortified with folic acid (30.7 ± 0.9 μg/100 g). The experimental pasta showed +44% folate higher than the average value found in wholemeal pasta.

Folate retention after cooking in our enriched pasta formulation was higher—68.5%, calculated on dry weight—compared to commercial whole-meal pasta (49.0%), and semolina pasta (56.2%), and remarkably outdid folate retention of fortified folic acid pasta (27.8% calculated on dry weight). A study conducted on 26 brands of noodles [54] reported folate losses vary from 6.1% to 30% in relation to different ingredients and cooking time. Similarly, average folate retention rate in unfortified wheat-based noodles was 78% in the study of Liang et al. [48]. In other starchy foods like potatoes, boiling for 60 min did not lead to a significant change in folate content (125.1 μg/100 g and 102.8 μg/100 g for raw and boiled potato respectively), even if skin was removed before [55].

Probably, the great stability of natural folate upon cooking could be given by the fact that these compounds are closely linked to macromolecules such as starch and protein, and therefore, far less likely to be lost in the cooking water.

Folate content of naturally enriched biscuits was quite interesting, 70.1 μg/100 g, about 5-times more than in commercial brand with the higher folate content (14.4 μg/100 g) and 10-times more than in brands with lower folate levels 6.2 μg/100 g. Notably, level of folate in new enriched experimental biscuits was slightly higher (+11.4%) than those found in fortified with folic acid biscuits 62.1 μg/100 g (range: 55.1 μg/100 g–69.0 μg/100 g).

Cookies fortified with folic acid (15 samples) analysed by Shakur et al. [56] showed a higher folate content of 94 ± 29 μg/100 g compared to our naturally enriched folate biscuits.

As regards other folic acid-fortified cereals, a research study was conducted [57] adding very high levels of folic acid (about 1 g/100 g) to wheat and corn flours, mixed and used for bakery products and couscous. Folic acid retention after cooking varied from 57% to 99%, with differences in relation to the cooking method used: dry heat method seems to be more conservative on folate content with respect to moist-heat method explained by the water-soluble nature of the vitamin. The highest folic acid retention was detected in corn products: authors [57] suggest that this is probably due to the interactions of the folic acid with the hydrophobic amino acids of Zein protein in corn. These results, although limited, highlight the need for other studies to identify best cereal ingredients, levels of folic acid or natural folate enrichment, and cooking method to obtain products with very high folate content and nutritional properties.

According to European Regulations (EC No 1924/2006 and EU No 1169/2011) for nutritional claim on label our experimental pasta and biscuits enriched in folate met the requirement for the nutritional claim “High folate” (>60 µg/100 g), as well as fortified pasta, whole meal pasta, and fortified commercial biscuits. It can be stated that the newly designed products can efficiently provide highly nutritious alternatives to meet the dietary requirements for folate. Obviously, the daily consumption of our new cereals products does not allow people to reach the recommended daily intake for folate both for adults (330 μg/100 g) and other population groups [20], and it will be fundamental to design other cereal products naturally enriched in folate (i.e., bread, breakfast, cereals, snack) and to promote the intake of other folate-rich food as legumes, vegetable, fruit and nuts.

## 4. Conclusions

This study showed that the barley pearling technique was effective at obtaining folate-rich flours. The pasta and biscuits newly designed in this study showed a very high folate content, especially if compared with the ones unfortified already on the market and with levels similar to folic acid-fortified foods (both cooked pasta and biscuits). In addition, the benefit of using barley by-products in the production of naturally folate-enriched foods stands also in the good content of bioactive compounds such as β-glucan.

Some other studies should be addressed to go more in depth about the difference of different folate forms of retention—folic acid and natural folate—in relation to the change of the physicochemical properties of food components during the preparation process e.g., mixing and cooking of ingredients in unfortified and fortified with folic acid cereals.

New products naturally enriched in folate could contribute to an improvement in folate nutritional status and decrease risk of congenital anomalies and cardiovascular and neurodegenerative diseases.

Indeed, this research lays some interesting foundations for the design of new folate and bioactive compound-enriched products that the consumers may appreciate for their organoleptic characteristics, natural folate enrichment and other functional properties as well.

## Figures and Tables

**Table 1 nutrients-14-03729-t001:** Formulations of experimental pasta and experimental biscuits with barley fraction enriched in folate.

Formulations	Ingredients
Folate enriched pasta	Commercial Semolina (Brand n 4): 61% Barley Fraction enriched in folate (BP2): 35% Gluten: 4%
Folate enriched biscuits	Commercial “0” flour (Brand n 4) (45%) and BP2 (55%) Sugar: 300 g Butter: 300 g Milk: 300 g Eggs: 150 g

**Table 2 nutrients-14-03729-t002:** Folate content in barley cultivars (Mean value ± s.d.).

Barley Cultivars	Total Folate Content (μg/100 g, d.m.)
Acquerelle	69.7 ± 7.4 ^b^
Bombay	64.9 ± 1.0 ^b^
Boreale	70.9 ± 17.3 ^abcd^
Braemar	75.5 ± 6.0 ^ab^
Calgary	61.2 ± 18.2 ^bcd^
Ceylon	65.8 ± 3.6 ^b^
Gothic	39.8 ± 2.7 ^e^
Kelibia	52.3 ± 2.2 ^d^
Ketos	53.2 ± 2.6 ^d^
Margaret	66.1 ± 13.4 ^bcd^
Messina	68.9 ± 2.8 ^b^
Metis	52.2 ± 3.5 ^d^
Natura	78.9 ± 7.3 ^a^
Otis	58.2 ± 2.5 ^cd^
Prestige	63.0 ± 10.2 ^bcd^
Rangoon	63.2 ± 17.2 ^abcd^
Sebastian	64.9 ± 7.9 ^bc^
Svenja	69.7 ± 7.4 ^b^
Ursa	58.7 ± 1.0 ^c^
Mean value Range CV%	63.0 39.8–78.9 14.6%

Different letters in the column indicate statistically significant differences (Kruskal–Wallis test, *p* < 0.05). CV = Coefficient of Variation

**Table 3 nutrients-14-03729-t003:** Yield, folate contents of pearling products from Natura cultivar (Mean value ± s.d).

Samples	Yield (%)	Total Folate (µg/100 g f.w.)
De-Hulled barley	100	69.1 ± 6.4
1st pearling		
Pearled Kernel (PK1)	85.7	54.1 ± 0.4
Pearling by-Products (BP1)	14.7	173.5 ± 2.2
2nd pearling		
Pearled Kernel (PK2)	80.7	30.3 ± 0.5
Pearling by-Products (BP2)	19.3	221.7 ± 7.0
3rd pearling		
Pearled Kernel (PK3)	68.4	20.0 ± 4.0
Pearling by-products (BP3)	31.6	177.3 ± 16.9

**Table 4 nutrients-14-03729-t004:** Folate content of commercial semolina and flours (Mean value ± s.d.).

Samples	Total Folate μg/100 g f.w.
Commercial semolina (Brand 1)	35.0 ± 1.0 ^a^
Commercial semolina (Brand 2)	35.3 ± 0.9 ^a^
Commercial semolina (Brand 3)	37.2 ± 1.2 ^a^
Commercial semolina (Brand 4)	40.2 ± 2.0 ^b^
Commercial 00 flour (Brand 1)	37.6 ± 1.2 ^a^
Commercial 00 flour (Brand 2)	43.3 ± 1.7 ^b^
Commercial 00 flour (Brand 3)	40.4 ± 0.9 ^b^
Commercial 00 flour (Brand 4)	58.2 ± 1.3 ^c^
Commercial 0 flour (Brand 1)	68.2 ± 2.0 ^e^
Commercial 0 flour (Brand 2)	50.2 ± 1.9 ^d^
Commercial 0 flour (Brand 3)	67.3 ± 1.7 ^e^
Commercial 0 flour (Brand 4)	69.3 ± 1.2 ^e^

Different letters in the column indicate statistically significant differences (Kruskal–Wallis test, *p* < 0.05).

**Table 5 nutrients-14-03729-t005:** Comparison between total folate content of enriched products and commercial products (Mean value ± s.d).

Cereal Products	Folate Content μg/100 g f.w. (Min–Max)	FR (%)
Enriched pasta uncooked	87.1 ± 3.8	68.5
Enriched pasta cooked	26.1 ± 1.1	
Commercial unfortified durum wheat pasta, uncooked (5 brands)	39.2 (36.7–44.3)	56.2
Commercial unfortified durum wheat pasta, cooked (5 brands)	12.3 (7.6–15.7)	
Commercial unfortified durum whole wheat pasta, uncooked (5 brands)	60.4 (50.5–61.1)	49.0
Commercial unfortified durum whole wheat pasta, cooked (5 brands)	16.5 (10.5–18.4)	
Commercial durum wheat pasta, fortified with folic acid, uncooked (1 brand)	200.9 ± 8.1	27.8
Commercial durum wheat pasta, fortified with folic acid, cooked (1 brand)	30.7 ± 0.9	
Enriched biscuits	70.1 ± 3.7	
Commercial unfortified biscuits (4 brands)	10.4 (6.2–14.4)	
Commercial fortified with folic acid biscuits (2 brands)	62.1 (55.1–69.0)	

FR, folate retention.

## Data Availability

The data presented in this study are available on request from the corresponding author.

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
