# Peer review of "Design of Cereal Products Naturally Enriched in Folate from Barley Pearling By-Products"

_nutrients, 2022, doi:10.3390/nu14183729_

Round 1

Reviewer 1 Report

The manuscript dealt with the enhancement of natural folate content, which is of a particular importance in countries where mandatory fortification is not practised and people are at risk of folate deficiency. The research question was sound and the results were promising. However, there were some points that warrant discussion, in addition to some minor comments.

Please pay attention to the use of folate/folates and the corresponding singular/plural forms throughout the manuscript. Table styles need to be unified. Avoid horisontal lines (Table 2; also totale -> total). Tables 3 & 4 could be combined. Table 5 belongs to the Materials and methods section. Table 6 should be modified, e.g. uncooked and cooked pasta of the same brand more close to each other. A professional language revision is recommended to improve the readability and fluency of the manuscript.

Other comments:

Abstract, Line 25: Without reading the manuscript, is is not clear that the retention means retention in cooking.

Introduction

There are several different font types. Folate status from the nutritional point of view is emphasised maybe too much – this part does not have to be so detailed when the aim was to study folate enhancement in cereal products.

Line 54: Check spelling: Chile

Materials and methods

2.1 Please provide information about the origin and storage of the cultivar samples.

2.2 Check punctuation. Please provide information about the sampling: how many packages were purchased, were the samples pooled etc.

2.3 “(…) underwent a three different pearling process(…)” A three-step process? Please clarify.

Were the pearling, pasta and bisquit making as well as pasta cooking performed only once?

2.9 The reference for enzyme treatments seems to be a conference abstract which is not easily available. Could you find another reference, or alternatively, provide some more details? How was the growth medium prepared?

Results and discussion

Please use past tense when reporting your results.

Lines 178-181: Something missing from the sentence – colon after Giordano et al.? In addition, folate levels…were detected.

Please discuss the variation among the studied cultivars. Had they been grown in the same place, same year? Were they obtained fresh? This information could/should also be presented in Materials and methods.

3.2 Rather folate content instead of concentration (also line 216)

Line 197: This sentence is not needed.

Lines 216-218: Please modify the units to allow comparison (ng/g -> ug/100 g)

Line 217: Mais -> maize

Lines 223-225: This sentence does not fit here.

Line 228: Please modify: ingredients very rich in folate

Lines 244-247: Not relevant here. Instead, what was the fortification level in the studied commercial products?

A suggestion: you could estimate the contribution of the new products to the daily reference intake values of folate in EU – can they be regarded as sources of folate or even rich in folate?

Reviewer 2 Report

 In view of the importance of the folic acid for health,  in the current original study the authors have tried to design new cereal based products – pasta and biscuits and compare to commercial samples of conventional and fortified cereal products. The paper is well written, the introduction justifies the aim of the study and in the discussion the authors comment on the effectiveness of their intervention and the significance of their results. Although the study is generally well designed, methodology needs improvement. The authors also mention the need for further studies.   

Comments

1.       Methods

a)Specify differences between “00” type and “0” type soft wheat flours because non experts are not familiar

b)Since trienzyme treatment was performed according to a method published in proceedings more details, i.e. selection of Ph, incubation time, accuracy and repeatability of the measurements, etc are necessary.

2.       Table 4: The phrase ‘Folate values with different letters differ significantly at P< 0.05’ needs more specifications

3.       Table 6: Show % change between each uncooked and cooked product.

4.       Limitations: Set any possible limitations

5.       References: Add last page where is missing

6.       English: minor editing is needed

Round 2

Reviewer 1 Report

The manuscript has been significantly improved. However, the part explaining the contribution of the improved products to the folate intake (lines 381-397) could be more concise. For instance, people do not usually eat uncooked pasta. Maybe just refer to the nutrition claims and not PRI?

Reviewer 2 Report

The authors have successfully answered to comments and the paper has been improved